# DIFF-ID: IDENTITY-CONSISTENT FACIAL IMAGE GENERATION AND MORPHING VIA DIFFUSION MODELS

## ABSTRACT

Generative diffusion models have revolutionized facial image synthesis, yet robust identity preservation in high-resolution outputs remains a critical challenge. This issue is especially vital for security systems, biometric authentication, and privacy-sensitive applications, where any drift in identity integrity can undermine trust and functionality. We introduce Diff-ID, a diffusion-based framework that enforces identity consistency while delivering photorealistic quality. Central to our approach is a custom 210K image dataset synthesized from CelebA-HQ, FFHQ, and LAION-Face and captioned via a fine-tuned BLIP model to bolster identity awareness during training. Diff-ID integrates ArcFace and CLIP embeddings through a dual cross-attention adapter within a fine-tuned Stable Diffusion U-Net. To further reinforce identity fidelity, we propose a pseudo-discriminator loss based on ArcFace cosine similarity with exponential timestep weighting. Experiments on held-out and unseen faces demonstrate that Diff-ID outperforms state-of-the-art methods in both identity retention and visual realism. Finally, we showcase a unified DDIM-based morphing pipeline that enables seamless facial interpolation without per-identity fine-tuning.

## 1 INTRODUCTION

Generative diffusion models have recently emerged as the leading paradigm for high-fidelity image synthesis, outperforming earlier techniques such as Generative Adversarial Networks (GANs) and Variational Autoencoders (VAEs) in training stability, diversity, and output quality Ho et al. (2020); Rombach et al. (2022). By iteratively refining noisy latent representations through a learned denoising process, diffusion-based frameworks can generate hyper-realistic images that capture intricate textures, complex structures, and subtle semantic variations.

The success of diffusion models spans diverse applications: from artistic content creation and virtual reality to medical imaging and scientific visualization, where photorealism and anatomical precision are critical Nichol & Dhariwal (2021); Saharia et al. (2022). In the domain of facial image synthesis, diffusion methods have demonstrated impressive visual quality, yet exhibit limitations when precise control over identity attributes is required. In particular, subtle identity drift manifesting as changes in facial geometry, expression, or distinguishing features can compromise the trustworthiness of generated outputs in scenarios demanding strict identity fidelity.

Maintaining robust identity consistency is a multifaceted challenge: models must preserve core identity-defining features (e.g., bone structure, eyes, and facial contours) under varying manipulations (e.g., expression, pose, lighting). However, most current diffusion methods Liu et al. (2023); Zhang et al. (2023b); Chen et al. (2023); Zhang et al. (2023a) emphasize global attribute changes, such as clothing styles or accessories, and produce avatar-like, non-photorealistic faces rather than fine-grained semantic edits or strict identity preservation. This identity control gap poses a significant barrier for security-sensitive applications, including biometric authentication, forensic analysis, and privacy-preserving data generation, where any deviation from true identity can have severe consequences. Synthetic morphing pipelines, used for adversarial robustness testing and privacy-aware dataset augmentation, further demand seamless identity interpolation without per-identity retraining or checkpoint swapping.

To address these challenges, we propose **Diff-ID**, a unified diffusion framework explicitly designed for the synthesis and morphing of identity-sensitive facial images. Our contributions include a **cus-**

**tom identity-centric dataset** of 210K images curated by blending and captioning CelebA-HQ, FFHQ, and LAION-Face with a fine-tuned BLIP model to ensure broad coverage of facial variations and semantic contexts; a **dual cross-attention adapter** that fuses identity embeddings from ArcFace with semantic embeddings from CLIP within a fine-tuned Stable Diffusion U-Net, enabling robust identity preservation while retaining fine-grained attribute control; a **pseudo-discriminator identity loss** that applies an ArcFace-based objective with exponentially increasing weight over diffusion timesteps to reinforce identity coherence throughout denoising; and a **unified morphing pipeline** that leverages DDIM inversion and joint embedding interpolation to produce smooth, identity-preserving morphs between arbitrary face pairs without additional fine-tuning or multiple checkpoints.

We validate Diff-ID on both held-out and unseen high-resolution facial datasets, showing marked improvements in ArcFace similarity scores and Fréchet Inception Distance (FID) compared to state-of-the-art baselines. Our approach sets a new benchmark for identity-aware diffusion models and paves the way for practical, security-driven applications in facial synthesis and biometric data augmentation.

## 2 RELATED WORK

Text-to-image diffusion models have rapidly become the dominant paradigm in generative image synthesis by combining iterative denoising processes with powerful text encoders. Early transformer-based approaches such as DALL·E Ramesh et al. (2021), CogView Ding et al. (2021), and Make-A-Scene Gal et al. (2022) demonstrated rich text–image correlations via discrete tokenization, but incurred substantial compute and latency costs at ultra–high resolutions due to quadratic self-attention complexity Ramesh et al. (2021).

Diffusion models overcome these bottlenecks by operating in continuous latent spaces and gradually transforming noise into coherent images. Song and Ermon formalized this process via stochastic differential equations, showing that score matching on Gaussian perturbations recovers complex data distributions Song & Ermon (2021). Subsequent advances improved both fidelity and efficiency. GLIDE proposed classifier-free guidance, allowing a simple trade-off between fidelity and diversity without an external classifier Nichol & Dhariwal (2021). DALL·E 2 adopted a hierarchical two-stage diffusion first by generating CLIP embeddings from text, then super-resolving back to the image space, which further increased sample quality Ramesh et al. (2022). The imagen achieved state-of-the-art FID by scaling the text encoders and tuning the noise schedules Saharia et al. (2022).

Latent Diffusion Models (LDMs) marked another leap by performing denoising in a compressed latent space, reducing memory and computation while retaining perceptual detail Rombach et al. (2022). Stable Diffusion, an open-source LDM at 512×512 resolution, democratized these techniques and inspired numerous extensions for style, structure, and spatial conditioning Zhang et al. (2023c). Despite these successes in text-driven generation, vanilla diffusion pipelines do not include mechanisms to anchor outputs to a specific face identity. As a result, attribute manipulations such as altering expression or pose can inadvertently shift identity features, leading to drift. This limitation emphasizes the need for identity-aware diffusion frameworks like our proposed Diff-ID.

**Zero-Shot Embedding Approaches** Zero-shot embedding techniques pushes pretrained diffusion models toward identity-preserving outputs without per-subject fine-tuning. Most approaches build on vision–language encoders such as CLIP Radford et al. (2021), injecting image or text embeddings into the generation process. IP-Adapter Liu et al. (2023) uses decoupled cross-attention to fuse visual prompts with text, while InstantID Chen et al. (2023) concatenates frozen face-recognition embeddings with text tokens at each denoising step. Although efficient, CLIP-based embeddings alone lack the fine-grained discriminability needed for strict identity fidelity. Diff-ID addresses this by integrating ArcFace embeddings Deng et al. (2019a), trained with angular-margin loss for inter-class separability, into a Stable Diffusion pipeline. By fusing ArcFace with CLIP features through a lightweight adapter, Diff-ID retains precise identity cues such as jawline geometry, eye proportions, nose shape, and lip contour.

**Decoupled Cross-Attention Mechanisms** Decoupled cross-attention separates text and image processing, improving semantic alignment between prompts and outputs. Methods like IP-

Adapter Liu et al. (2023) and InstantID Chen et al. (2023) employ this strategy to refine attribute consistency, but reliance on CLIP embeddings alone often leads to identity drift. Diff-ID shows that carefully tuning existing cross-attention layers is sufficient for both identity preservation and realism, avoiding the architectural overhead of decoupled modules while still allowing for extensions such as ControlNet when fine-grained semantic control is required.

**Stacked-ID Embeddings for Identity Fidelity** PhotoMaker Zhang et al. (2023b) exemplifies stacked-ID approaches, encoding multiple reference images into a combined embedding that maintains consistency across pose, expression, and lighting. While effective, this strategy incurs significant computational cost, requiring custom-trained encoders and large memory overhead. Diff-ID sidesteps these limitations by leveraging off-the-shelf ArcFace embeddings Deng et al. (2019a) within a Stable Diffusion backbone Rombach et al. (2022), achieving comparable identity fidelity to stacked-ID methods with reduced complexity and latency.

**Diffusion-Based Morphing and Synthetic Identity Generation** GAN-based morphing methods such as StyleGAN Karras et al. (2019), StyleFlow Abdal et al. (2021), and GANSpace Härkönen et al. (2020) demonstrate smooth latent-space interpolations but suffer from mode collapse and checkpoint dependency. Diffusion alternatives like DiffMorpher Lee et al. (2024) invert images into DDIM latents and blend them, but require per-identity LoRA Hu et al. (2021) fine-tuning and multiple checkpoints. In contrast, Diff-ID performs morphing natively through DDIM inversion combined with dual cross-attention: ArcFace embeddings preserve fine identity cues while CLIP embeddings guide semantic consistency. This unified design enables smooth, photorealistic interpolations without fine-tuning or checkpoint swapping, supporting continuous morphing, synthetic ID generation, privacy-preserving blending, and adversarial robustness testing.

## 3 METHODOLOGY

### 3.1 DATA PREPARATION AND CURATION

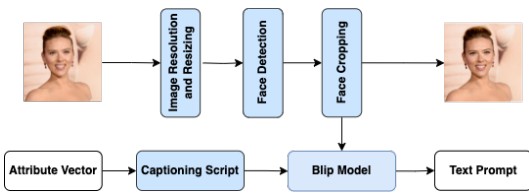

Figure 1: Data pipeline

We curated a large-scale dataset composed of images from three well-known high-resolution facial datasets: CelebA-HQ, FFHQ, and LAION-Face. All images were resized to a uniform resolution of 512 to ensure consistency during model training. Unlike CelebA-HQ, which provides 40 annotated binary attributes per image, FFHQ and LAION-Face lacked descriptive captions. To address this, we developed a custom script that converts CelebA-HQ's attribute vectors into structured sentence-style captions. We then fine-tuned a BLIP model on this annotated subset to generate high-quality captions for FFHQ and LAION-Face images. To maintain data quality and semantic consistency, we removed images with occlusions (e.g., sunglasses, hats), low-quality images, and any samples undetectable by InsightFace Library Ren et al. (2023); Guo et al. (2021); Gecer et al. (2021); An et al. (2022); Deng et al. (2019b) . This rigorous cleaning and captioning process resulted in a dataset of approximately 210,000 images, each paired with a descriptive and unique caption, enabling more expressive and attribute-guided generation while maintaining identity fidelity.

### 3.2 MODEL ARCHITECTURE

#### 3.2.1 EMBEDDING EXTRACTION AND FUSION

In many prior models, identity features are extracted using a single embedding source either via ArcFace (e.g. in Arc2Face) or through CLIP's image encoder (e.g. in IPAdapter and Instant ID).

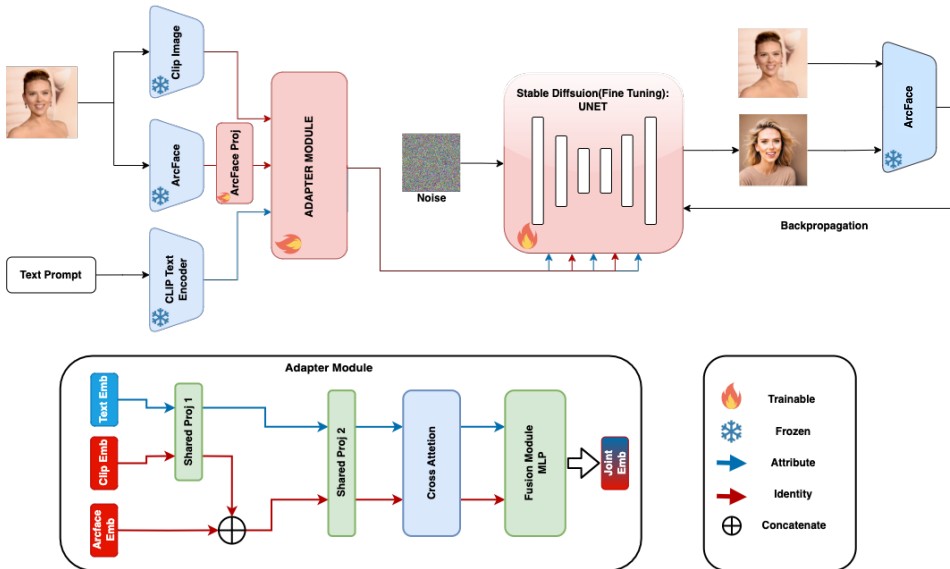

Figure 2: The pipeline begins by extracting semantic features from the CLIP encoder and identity features from the ArcFace model, which are then projected and fused to form a unified identity representation. This fused representation is aligned with the CLIP text embedding via a shared projection layer. Next, the aligned embeddings are processed through a dual cross-attention mechanism that separately enriches identity and attribute information. The outputs of this module are concatenated and passed through a Fusion MLP to generate a refined embedding. Finally, we cross-attend these joint embeddings into the pre-trained Stable Diffusion 1.5 U-Net, guiding the diffusion process to produce high-fidelity images with robust identity preservation.

However, relying solely on one modality can be limiting. ArcFace is highly effective at capturing fine-grained, identity-specific details crucial for face recognition, but it may not fully capture the semantic nuances necessary for aligning with textual attributes. In contrast, CLIP's image encoder provides robust semantic representations that align well with text inputs, yet it may overlook subtle identity cues essential for preserving individual characteristics. To address these complementary strengths and weaknesses, our **Diff-ID** model integrates both embedding types, thereby ensuring a more comprehensive representation of identity and semantics.

Let $A$ denote the input attribute text prompt and $I$ the reference image. The text embedding is computed as $e_t = \text{CLIP}_{\text{text}}(A)$ while the image embedding is $e_i = \text{CLIP}_{\text{image}}(I)$. Here, $\text{CLIP}_{\text{text}}$ and $\text{CLIP}_{\text{image}}$ represent the text and image encoding components of CLIP, respectively, $e_t$ encapsulating the semantic content of the prompt and $e_i$ providing complementary visual context. Given the same input image $I$, the ArcFace model produces a 512-dimensional identity embedding $e_{\text{arc}} = \text{ArcFace}(I)$. This embedding captures unique identity-specific features that are critical for maintaining identity fidelity during attribute manipulation.

**Embedding Projection and Fusion** To integrate multi-modal information, we first project the ArcFace and CLIP image embeddings into a common latent space. Their projected forms are given by $e'_{\text{arc}} = W_{\text{arc}} e_{\text{arc}}$ and $e'_i = W_i e_i$, where $W_{\text{arc}}$ and $W_i$ are learnable projection matrices. Next, we concatenate these projected embeddings to form a combined image representation, $e_{\text{img}} = \text{Concat}(e'_{\text{arc}}, e'_i)$. To further capture salient identity features, we apply both max pooling and average pooling on $e_{\text{img}}$, yielding $e_{\text{max}} = \text{MaxPool}(e_{\text{img}})$ and $e_{\text{avg}} = \text{AvgPool}(e_{\text{img}})$. We then form the final identity representation by concatenating the individual components $e_{\text{id}} = \text{Concat}(e'_{\text{arc}}, e'_i, e_{\text{max}}, e_{\text{avg}})$. In parallel, the CLIP text embedding $e_t$ is aligned with the identity modality by passing both $e_{\text{id}}$ and $e_t$ through a shared linear projection with weight matrix $W_s$, resulting in $\tilde{e}_{\text{id}} = W_s e_{\text{id}}$ and $\tilde{e}_t = W_s e_t$. This yields two separate yet aligned branches, the identity branch and the attribute (text) branch, for subsequent processing.

### 3.2.2 DUAL CROSS-ATTENTION MECHANISM

To effectively integrate and refine the identity and attribute information, we propose a dual cross-attention mechanism. The motivation behind this design is twofold. First, by enabling bidirectional interactions, the model allows each modality to inform and enhance the other ensuring that identity features are enriched by semantic cues from the text and vice versa. Second, by omitting softmax normalization, the mechanism preserves the raw magnitude relationships between features, which can help retain fine-grained details that might otherwise be diminished.

We compute two cross attention branches jointly: in the identity branch the identity embedding $\tilde{e}_{\text{id}}$ serves as the query and the CLIP text embedding $\tilde{e}_t$ provides both key and value, while in the attribute branch $\tilde{e}_t$ is the query and $\tilde{e}_{\text{id}}$ provides key and value; both branches follow

$$e = \left( \frac{QK^\top}{\sqrt{d}} \right) V. \tag{1}$$

For the identity branch we set $Q_{\text{id}} = W_q^{\text{id}} \tilde{e}_{\text{id}}, K_{\text{id}} = W_k^{\text{id}} \tilde{e}_t, V_{\text{id}} = W_v^{\text{id}} \tilde{e}_t$, for the attribute branch we set $Q_{\text{attr}} = W_q^{\text{attr}} \tilde{e}_t, K_{\text{attr}} = W_k^{\text{attr}} \tilde{e}_{\text{id}}, V_{\text{attr}} = W_v^{\text{attr}} \tilde{e}_{\text{id}}$, where $d$ is the embedding dimensionality.

### 3.2.3 FUSION MLP

Although using an MLP for feature fusion is standard practice, our design incorporates a Fusion MLP Module for an important reason: it enables the network to learn complex, nonlinear interactions between the enriched identity and attribute features. In doing so, we refine the joint representation in a way that preserves identity specific details while effectively guiding identity specific attributes. The outputs from the dual cross attention modules, the enriched identity representation $e_{\text{identity}}$ and the enriched attribute representation $e_{\text{attribute}}$, are concatenated to form a unified feature representation $e_{\text{fused}} = \text{Concat}(e_{\text{identity}}, e_{\text{attribute}})$. This fused embedding is then processed through a Fusion MLP block to generate a refined embedding $e_{\text{refined}} = \text{MLP}(e_{\text{fused}})$, where the MLP is defined as $\text{MLP}(x) = \sigma\big(W_2 \, \sigma(W_1 x + b_1) + b_2\big)$. Here, $W_1$ and $W_2$ are learnable weight matrices, $b_1$ and $b_2$ are biases, and $\sigma$ denotes the ReLU activation function.

### 3.3 OBJECTIVE FUNCTION

Our objective function is carefully designed to balance two essential goals: high-quality denoising and robust identity preservation. The denoising loss, adapted from the Stable Diffusion framework, drives the model to effectively remove noise and reconstruct images that follow the target distribution. In parallel, the adversarial identity loss, computed using ArcFace embeddings, ensures that the generated images retain core identity features. Recognizing that identity details are less discernible at higher noise levels, we incorporate an exponential dynamic weighting strategy. This strategy modulates the importance of identity loss throughout the diffusion process, emphasizing identity preservation when the noise is lower and identity information is more reliable.

**Denoising Loss** Following the Stable Diffusion framework, the model minimizes the difference between the true noise $\epsilon$ added during the forward process and the noise predicted by the network $\epsilon_\theta(z_t, t)$. The denoising loss is given by the equations in Appendix A.1.4

**Adversarial Identity Loss** To ensure that generated images preserve core identity features, ArcFace is employed as a pseudo-discriminator. It extracts identity embeddings $e_{\text{original}}$ and $e_{\text{generated}}$ from the original and generated images, respectively. The identity loss is defined as

$$\mathcal{L}_{\text{identity}} = 1 - \text{Cosine Similarity}(e_{\text{original}}, e_{\text{generated}}) \tag{2}$$

**Exponential Dynamic Weighting Strategy** Since identity features become less discernible at higher noise levels, we weight the identity loss exponentially based on the current timestep. Let $\tau = \frac{t}{T}$ (with $t$ the current timestep and $T$ the total number of timesteps) and define $W_t = \exp(-k\,\tau)$, with $k$ a decay hyperparameter controlling the rate of exponential decrease as seen in Fig. 5 in Appendix A.1.4. The weighted identity loss per sample is then $\mathcal{L}_{\text{identity, weighted}} = \mathbb{E}[W_t \cdot \mathcal{L}_{\text{identity}}]$.

**Overall Training Objective**   The final training loss is a combination of the denoising loss and the dynamically weighted identity loss:

$$\mathcal{L}_{\text{total}} = \mathcal{L}_{\text{denoise}} + \lambda_{\text{identity}} \cdot \mathcal{L}_{\text{identity, weighted}} \tag{3}$$

where $\lambda_{\text{identity}} = 0.10$ is a hyperparameter balancing the importance of identity preservation against denoising accuracy.

### 3.4 Morphology-Based Generation

We extend Diff-ID to support identity-preserving facial morphing between two source identities without requiring per-identity fine-tuning. Our method builds on the pretrained dual-branch adapter and U-Net backbone and leverages deterministic sampling via DDIM to ensure stable and high-fidelity outputs. The generation architecture as seen in Fig. 6 in Appendix A.1.5 proceeds in three key stages:

Given two real input images $x_1$ and $x_2$, we use DDIM inversion Song & Ermon (2020) to recover their corresponding noise latents $z_1^T$ and $z_2^T$ at the final timestep $T$ of the diffusion process. This inversion is defined in the Appendix A.1.5. This step ensures the latent noise preserves both the visual and identity information of each input image in the diffusion space.

We extract identity embeddings $e_1$ and $e_2$ using the pretrained CLIP and ArcFace encoders, as integrated into Diff-ID's dual-adapter module. To morph between the two identities, we perform spherical linear interpolation (SLERP) of the embeddings on the unit hypersphere as seen in the Appendix A.1.5. In parallel, we interpolate the latent noise vectors $z_1^T$ and $z_2^T$ using linear interpolation as derived in the Appendix A.1.5 ensuring the structure and fine details of both identities are blended in the denoising trajectory. Combining both embedding and latent interpolation allows for smooth transitions in both semantic identity space and low-level generative signal.

Using the interpolated noise vector $z_{\text{mix}}^T$ and the blended identity embedding $e_{\text{mix}}$, we perform deterministic DDIM sampling guided by Diff-ID's dual cross-attention mechanism producing the final morphed image $\hat{x}$ that exhibits characteristics from both source identities while preserving realism and coherence. Our approach supports continuous control over the morph factor $\alpha$, enabling a spectrum of identities between $x_1$ and $x_2$. Unlike other methods requiring multiple fine-tuned checkpoints (DreamBooth Ru et al. (2022) or DiffMorpher Lee et al. (2024)) or low-rank adaptation layers such as LoRA Hu et al. (2021), our pipeline operates in a single unified model.

### 3.5 Training Configuration and Hyperparameters

The training setup for Diff-ID involves a mix of frozen and trainable layers, with the VAE encoder and decoder kept frozen to preserve the latent to image transformations without additional fine tuning Rombach et al. (2022), and both CLIP and ArcFace left frozen to retain their pretrained strengths for semantic and identity specific features Deng et al. (2019a). Diff-ID is trained with the Adam optimizer at a learning rate of $1 \times 10^{-5}$ and a batch size of 16 using mixed precision (fp16). Gradient checkpointing is enabled to control memory consumption and allow larger effective batches. Training is run for 1,000,000 steps on $4\times$ NVIDIA A100 80GB GPUs, reaching stable performance in approximately 72 hours.

## 4 Evaluation Framework and Results

### 4.1 Qualitative and Quantitative Framework

Following established practice Liu et al. (2023); Zhang et al. (2023a), we present side by side image grids comparing Diff ID outputs with baseline methods. This visual inspection emphasizes **identity fidelity**, i.e., the extent to which characteristic facial features are retained, and **perceptual realism**, i.e., the absence of unnatural artifacts or distortions. By highlighting both successes and failure modes, these qualitative examples illustrate how Diff ID balances identity consistency against creative variation. Complementing the qualitative analysis, we evaluate two complementary dimensions quantitatively. For **identity preservation**, we compute cosine similarity between 512 dimensional embeddings extracted by a pretrained ArcFace model Deng et al. (2019a) for each

source–generated pair; ArcFace is optimized for face recognition and serves as a reliable proxy for identity fidelity, though it may overestimate similarity for highly stylized outputs. For **visual realism**, we report Fréchet Inception Distance (FID) Heusel et al. (2017), which measures the divergence between distributions of generated and real images in the Inception v3 feature space; while FID captures large scale discrepancies, it can under represent small yet perceptually important facial artifacts. To jointly assess identity preservation and realism, we introduce the **Face Image Quality (FIQ Score)** as a normalized ratio of Face Similarity (FS%) to FID,

$$\text{FIQ Score} = \frac{100 \cdot \text{FS\%}}{\text{FID}}, \tag{4}$$

which establishes a dependent relationship where higher Face Similarity increases the score and higher FID decreases it, thereby penalizing unrealistic outputs. This integrated measure, together with the qualitative analysis, provides a balanced and interpretable evaluation of generative face models by explicitly capturing the interplay between identity fidelity and visual realism.

## 4.2 RESULTS

| | Orignal | IPAdapter | PhotoMaker | InstantID | Arc2Face | Ours |
|---|---|---|---|---|---|---|

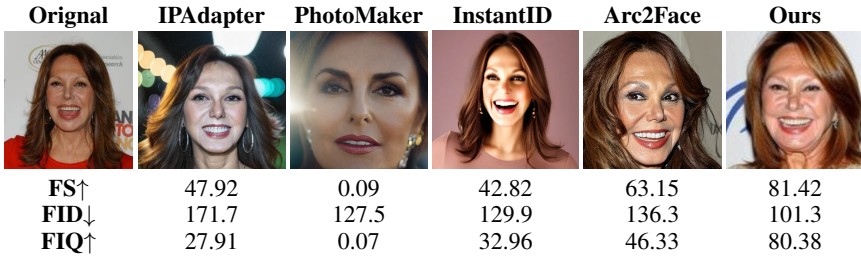

| | | | | | | |
|---|---|---|---|---|---|---|
| **FS↑** | | 47.92 | 0.09 | 42.82 | 63.15 | 81.42 |
| **FID↓** | | 171.7 | 127.5 | 129.9 | 136.3 | 101.3 |
| **FIQ↑** | | 27.91 | 0.07 | 32.96 | 46.33 | 80.38 |

Figure 3: Comparison on Identity 1 across state of the art and our method.

### 4.2.1 IDENTITY PRESERVATION

Our comparative image grid in Fig. 3 and the extended results in Appendix A.2, Fig. 7, show that Diff-ID preserves critical identity-specific details such as facial bone structure, eye spacing, nose shape, and lip contour more faithfully than competing methods. IPAdapter Liu et al. (2023) and PhotoMaker Zhang et al. (2023b) exhibit the weakest preservation, with noticeable drift in geometric structure and texture. Arc2Face Papantoniou et al. (2024) maintains stronger identity cues but suffers from reduced sharpness and consistency across samples. Instant ID Chen et al. (2023) achieves the highest raw Face Similarity (FS%) on both validation (75.13) and unseen data (74.12) as seen in Table 1, yet its outputs often lack finer detail and can overemphasize stylized similarities. Diff-ID ranks closely behind in FS% (72.68 / 71.53) but does so while maintaining much stronger visual fidelity. This balance becomes evident when considering the Face Image Quality (FIQ) score, which integrates realism into the evaluation. Here, Diff-ID outperforms all baselines, reaching 71.74 on validation and 69.32 on unseen data, indicating that our model preserves identity in a way that is both consistent and photorealistic.

### 4.2.2 PERCEPTUAL REALISM

Perceptual realism is measured by Fréchet Inception Distance (FID) and incorporated into the FIQ Score. Table 1 shows that Diff-ID attains the lowest FID across both validation (101.31) and unseen splits (103.19), outperforming IPAdapter, PhotoMaker, Instant ID, and Arc2Face Papantoniou et al. (2024). This directly translates into the highest overall FIQ scores, despite a modest drop in FS% compared to Instant ID Chen et al. (2023). In contrast, PhotoMaker Zhang et al. (2023b) achieves relatively low FID (127.47 / 113.50) but fails to preserve identity well, leading to weak FS and FIQ. Arc2Face Papantoniou et al. (2024) achieves a balanced mid-range performance, with solid FS but higher FID than Diff-ID. These trends highlight that Diff-ID not only anchors identity more reliably than most baselines but also produces coherent textures, skin tones, and background details that avoid the cartoonish or over-smoothed artifacts present in others. We note that FID may overlook perceptually salient flaws, while FS% can over-reward stylized similarity. Nonetheless, Diff-ID's superior FIQ demonstrates that it provides the most reliable trade-off between photorealism and identity retention.

Table 1: Evaluation on validation and unseen sets

| Model | Validation Set | | | Unseen Data | | |
|---|---|---|---|---|---|---|
| | FS↑ | FID↓ | FIQ↑ | FS↑ | FID↓ | FIQ↑ |
| IP-Adapter Liu et al. (2023) | 40.53 | 171.71 | 23.60 | 35.90 | 151.56 | 23.69 |
| Photomaker Zhang et al. (2023b) | 33.87 | 127.47 | 26.57 | 29.56 | 113.50 | 26.04 |
| Instant ID Chen et al. (2023) | **75.13** | 129.98 | 57.80 | **74.12** | 119.39 | 62.08 |
| Arc2Face Papantoniou et al. (2024) | 73.71 | 136.25 | 54.10 | 72.51 | 110.69 | 65.51 |
| Diff-ID (Ours) | 72.68 | **101.31** | **71.74** | 71.53 | **103.19** | **69.32** |

## 5 ABLATION STUDY

### 5.1 MORPHING

Figure 8 in Appendix A.3.1 illustrates three morphing strategies. Unlike conventional pixel-space morphing, our approach first maps the original faces into the latent diffusion space via DDIM inversion; morphing is then performed directly on these latent identities, ensuring face-specific and semantically consistent transitions. The strategies shown are: (i) embedding-only interpolation of ArcFace identity vectors, (ii) Diff-ID morph via linear interpolation (Lerp), and (iii) Diff-ID morph via spherical interpolation (Slerp).

Embedding-only interpolation (top row) produces overly smooth, low-detail transitions that often misalign key facial features and lack high-frequency texture. In contrast, Diff-ID Lerp (middle row) injects identity and semantic cues at each denoising step, producing sharp, coherent blends, although some intermediate frames still exhibit mild 'ghosting' of features. Diff-ID Slerp (bottom row) further regularizes interpolation on the hypersphere, improving midpoint consistency in bone structure, skin texture, and overall likeness. Although we do not report quantitative ArcFace scores or FIQ for each frame here, the qualitative visual fidelity of Diff-ID Slerp aligns with its superior aggregate FIQ in our main evaluation: it strikes the best balance between identity retention and perceptual realism. We acknowledge that pure FS% on individual frames can be inflated by stylized or avatar-like artifacts; instead, these visual results confirm that our diffusion-based joint-interpolation approach produces smooth, photorealistic face morphs without the need for per-identity fine-tuning or multiple checkpoints.

### 5.2 IMPACT OF DIFFERENT ADAPTER STRATEGIES

Table 2: Adapter Comparative Analysis

| Model Variant | Face Similarity |
|---|---|
| DiffID-ArcFace | 48.91 |
| DiffID-CLIP | 53.41 |
| DiffID-Joint | 65.83 |
| DiffID-Final | **72.68** |

Qualitatively, Fig. 9 in the Appendix A.3.2 shows that the ArcFace-only adapter (DiffID-ArcFace) preserves broad geometry (jawline, eye spacing) but lacks fine texture detail. The CLIP-only adapter (DiffID-CLIP) recovers richer semantic context and sharper textures but introduces identity drift-nose shape and lip contour deviate from the source. The concatenated Joint adapter DiffID-Joint) balances geometry and texture but still shows slight mid-face averaging. Our final DiffID-Final adapter robustly reproduces high-frequency details (pores, subtle wrinkles) and faithfully maintains bone structure, lip shape, and eye geometry across examples. Quantitively, Table 2 shows that the ArcFace-only adapter scores 48.91%, reflecting limited textural fidelity. The CLIP-only variant achieves 53.41% by leveraging semantic detail but misaligns fine identity cues. The combined Joint adapter jumps to 65.83%, highlighting the benefit of multi-modal fusion. Finally, our DiffID-Final configuration reaches 72.68%, validating how dual cross-attention and the Fusion MLP synergistically maximize identity coherence without sacrificing realism. These ablations confirm that both ArcFace and CLIP embeddings are necessary: ArcFace provides precise identity discrimination, while CLIP enriches semantic and textural context. Their joint integration in our adapter is critical for achieving state-of-the-art identity retention in diffusion-based facial synthesis.

## 5.3 IMPACT OF IDENTITY LOSS

Figure 10 in Appendix A.3.3 compares training curves for the denoising loss when using our pseudo-discriminator identity loss versus a baseline without it. Incorporating the identity term yields faster early convergence: the model quickly learns to preserve coarse identity features and minimizes the combined loss, whereas the baseline drifts more gradually as it struggles to infer identity from reconstruction alone. By emphasizing identity coherence at low noise levels, our weighted identity loss guides the network toward stable identity retention from the outset.

Figure 4 further illustrates the impact of varying the identity-loss weight $\lambda$. At low values ($\lambda = 0.10$), the identity term gently steers the diffusion process, yielding high-fidelity identity retention without disturbing the denoising dynamics. As $\lambda$ increases to 0.30–0.50, the model over-prioritizes identity, causing artifacts and degraded denoising quality. Therefore, features become unnaturally sharp or "stuck," and background details collapse. These results confirm that a moderate weighting (around 0.10–0.20) achieves the best balance between identity fidelity and photorealism, validating our choice of $\lambda = 0.10$ in Eq. 3 of Section 3.3.

| Original | $\lambda = 0.10$ | $\lambda = 0.20$ | $\lambda = 0.30$ | $\lambda = 0.40$ | $\lambda = 0.50$ |

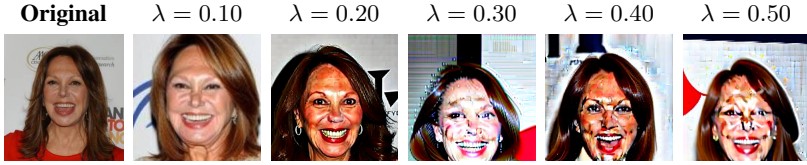

Figure 4: Effect of identity regularization strength ($\lambda$) on generated outputs.

## 6 LIMITATIONS AND FUTURE WORK

While Diff-ID sets a new standard for identity-consistent face synthesis and morphing, several limitations remain. First, by relying on frozen CLIP image and text encoders for semantic context, our framework lacks explicit control over non-facial elements, which remain governed by coarse CLIP embeddings rather than dedicated spatial or attribute-specific modules. Second, Diff-ID is optimized for single, frontal face crops; extreme head poses, severe occlusions, or full-body scenes can degrade identity fidelity, since the model has not been trained to disentangle complex scene elements or non-facial regions. Third, our morphing pipeline employs simple linear and spherical interpolation of latents and identity embeddings, which, while effective for smooth transitions, does not support attribute-conditioned or region-specific morph trajectories (e.g., selectively blending expressions or hairstyles).

Future research can address these gaps by integrating ControlNet-style spatial conditioning or learned attention masks to separately modulate background and clothing attributes. Enhancing the dataset with multi-view and occluded face samples, or incorporating 3D-aware diffusion priors, could improve robustness to pose and occlusion. On the morphing side, developing learned interpolation networks or diffusion paths guided by semantic anchors would enable finer-grained, attribute-aware transitions. By extending Diff-ID in these directions, one can build a more generalizable and controllable platform for secure, identity-anchored image synthesis and morphing.

## 7 CONCLUSION

In this work, we introduced **Diff-ID**, a unified diffusion framework for high-resolution facial image generation and morphing that explicitly enforces identity consistency. By fusing ArcFace and CLIP embeddings through a lightweight dual cross-attention adapter within a fine-tuned Stable Diffusion U-Net, and by incorporating a pseudo-discriminator identity loss with exponential timestep weighting, Diff-ID achieves state-of-the-art performance in both identity retention and visual realism. Our extensive evaluations on held-out and unseen face sets demonstrate superior ArcFace similarity scores and Face Iamge Quality metrics compared to leading baselines. Furthermore, our DDIM-based morphing pipeline delivers smooth, photorealistic face interpolations without per-subject fine-tuning or multiple checkpoints. Diff-ID lays the groundwork for practical applications in biometric security, privacy-preserving data augmentation, and photorealistic avatar creation.

## 8 Reproducibility And Ethics Statement

To ensure reproducibility, we will release the Diff-ID codebase and other scripts on our github repository in due time. All experiments were conducted on publicly available datasets (CelebA-HQ, FFHQ, and LAION-Face). Our identity-captioning pipeline and curated splits will be shared upon reasonable request, subject to ethical review, given the sensitivity of identity-specific facial generation. We acknowledge the potential for misuse such as deepfakes generation and therefore adopt a responsible-release policy that includes usage terms and safeguards. At the same time, we believe rigorous, transparent research on identity-preserving generation and morphing is essential for developing countermeasures and strengthening content authenticity tools. Understanding how such systems work is prerequisite to detecting, watermarking, and mitigating their abuse.

## 9 Use of Large Language Models (LLMs)

We clarify that Large Language Models (LLMs) were not used to generate any scientific content, ideas, or explanations in this work. Their role was limited strictly to supporting grammar, spelling, and sentence-structure refinement, with the goal of improving readability and consistency for the reader. Additionally, LLMs were employed to check and standardize formatting across sections. All conceptual contributions, methodologies, experiments, and analyses reported in this paper are the result of the authors' original work.

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

# A APPENDIX

## A.1 FURTHER METHODOLOGICAL DISCUSSION

This section collects the detailed mathematical formulations referenced in the main paper. Each equation is presented with a short explanation for clarity.

### A.1.1 EMBEDDING EXTRACTION AND FUSION.

ArcFace and CLIP image embeddings are projected into a common space:

$$e'_{\text{arc}} = W_{\text{arc}} e_{\text{arc}}, \qquad e'_i = W_i e_i, \tag{5}$$

A combined identity embedding is then formed by concatenating the projected embeddings along with pooled variants:

$$e_{\text{id}} = \text{Concat}\big(e'_{\text{arc}}, e'_i, e_{\text{max}}, e_{\text{avg}}\big), \tag{6}$$

where $e_{\text{max}} = \text{MaxPool}(e_{\text{img}})$ and $e_{\text{avg}} = \text{AvgPool}(e_{\text{img}})$. To align with text, both the identity and text embeddings are projected through a shared transformation:

$$\tilde{e}_{\text{id}} = W_s e_{\text{id}}, \qquad \tilde{e}_t = W_s e_t, \tag{7}$$

### A.1.2 DUAL CROSS ATTENTION.

The generic cross-attention template is

$$e = \left(\frac{QK^\top}{\sqrt{d}}\right) V, \tag{8}$$

For the identity branch:

$$Q_{\text{id}} = W_q^{\text{id}} \tilde{e}_{\text{id}}, \quad K_{\text{id}} = W_k^{\text{id}} \tilde{e}_t, \quad V_{\text{id}} = W_v^{\text{id}} \tilde{e}_t, \tag{9}$$

$$e_{\text{identity}} = \left(\frac{Q_{\text{id}} K_{\text{id}}^\top}{\sqrt{d}}\right) V_{\text{id}}, \tag{10}$$

For the attribute branch:

$$Q_{\text{attr}} = W_q^{\text{attr}} \tilde{e}_t, \quad K_{\text{attr}} = W_k^{\text{attr}} \tilde{e}_{\text{id}}, \quad V_{\text{attr}} = W_v^{\text{attr}} \tilde{e}_{\text{id}}, \tag{11}$$

$$e_{\text{attribute}} = \left(\frac{Q_{\text{attr}} K_{\text{attr}}^\top}{\sqrt{d}}\right) V_{\text{attr}}, \tag{12}$$

### A.1.3 FUSION MLP.

The two enriched embeddings are concatenated and refined via a multi-layer perceptron:

$$e_{\text{fused}} = \text{Concat}(e_{\text{identity}}, e_{\text{attribute}}), \tag{13}$$

$$e_{\text{refined}} = \text{MLP}(e_{\text{fused}}), \tag{14}$$

$$\text{MLP}(x) = \sigma\big(W_2 \, \sigma(W_1 x + b_1) + b_2\big), \tag{15}$$

where $\sigma$ denotes ReLU.

### A.1.4 TRAINING OBJECTIVES.

The denoising loss follows the Stable Diffusion framework:

$$\mathcal{L}_{\text{denoise}} = \mathbb{E}_{z_0, t, \epsilon} \big[\|\epsilon - \epsilon_\theta(z_t, t)\|_2^2\big], \tag{16}$$

$$z_t = \sqrt{\alpha_t} \, z_0 + \sqrt{1 - \alpha_t} \, \epsilon, \qquad \epsilon \sim \mathcal{N}(0, I), \tag{17}$$

Identity preservation is enforced via cosine similarity:

$$\mathcal{L}_{\text{identity}} = 1 - \text{CosineSim}(e_{\text{original}}, e_{\text{generated}}), \tag{18}$$

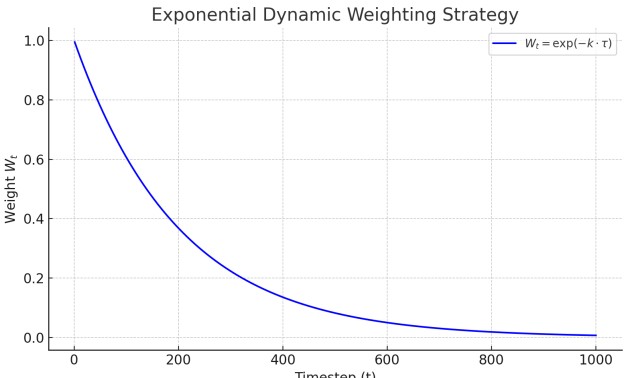

Figure 5: Weight-decay curve over timesteps with $k = 5$.

To adapt over timesteps, exponential weighting is applied:

$$W_t = \exp(-k\tfrac{t}{T}), \tag{19}$$

$$\mathcal{L}_{\text{identity,weighted}} = \mathbb{E}\big[W_t \cdot \mathcal{L}_{\text{identity}}\big]. \tag{20}$$

The total training objective is:

$$\mathcal{L}_{\text{total}} = \mathcal{L}_{\text{denoise}} + \lambda_{\text{identity}} \cdot \mathcal{L}_{\text{identity,weighted}}, \tag{21}$$

### A.1.5 MORPHOLOGY-BASED GENERATION.

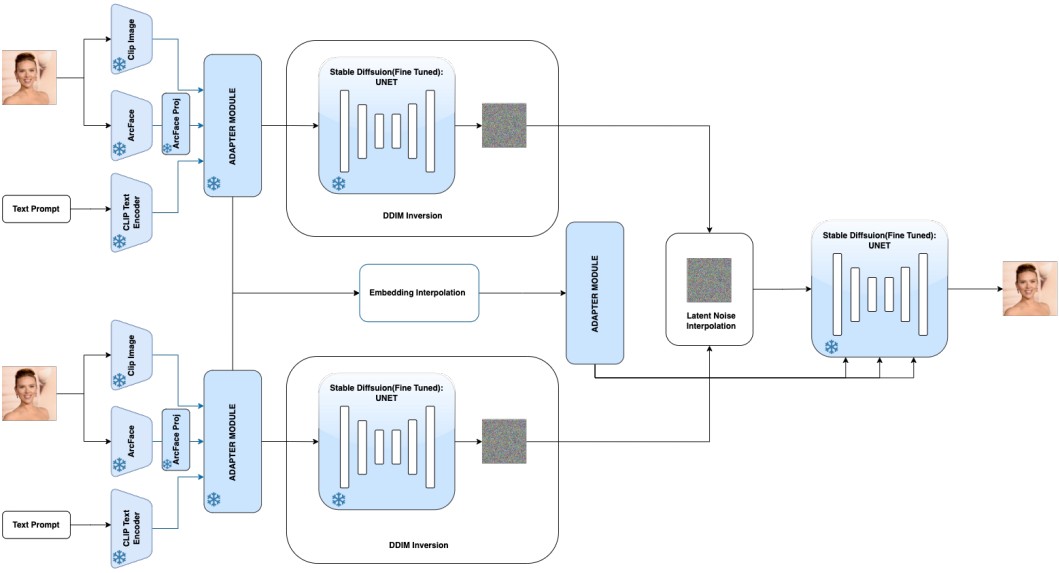

Figure 6: DiffID-Morph: Our identity-preserving facial morphing pipeline built on DDIM inversion and dual-embedding interpolation.

DDIM inversion recovers latent noise for an input image:

$$z^T = \Phi_\theta^{-1}(x), \tag{22}$$

Identity embeddings are interpolated with SLERP:

$$e_{\text{mix}} = \frac{\sin((1-\alpha)\omega)}{\sin\omega}\, e_1 + \frac{\sin(\alpha\omega)}{\sin\omega}\, e_2, \qquad \omega = \arccos\left(\frac{e_1 \cdot e_2}{\|e_1\|\|e_2\|}\right), \tag{23}$$

Noisy latents are linearly interpolated:

$$z_{\text{mix}}^T = (1 - \alpha)z_1^T + \alpha z_2^T, \tag{24}$$

Finally, deterministic DDIM sampling yields the morphed image:

$$\hat{x} = \text{DDIMSample}(z_{\text{mix}}^T; e_{\text{mix}}, \theta), \tag{25}$$

## A.2   FURTHER QUALITATIVE RESULTS

Additional qualitative results are provided here.

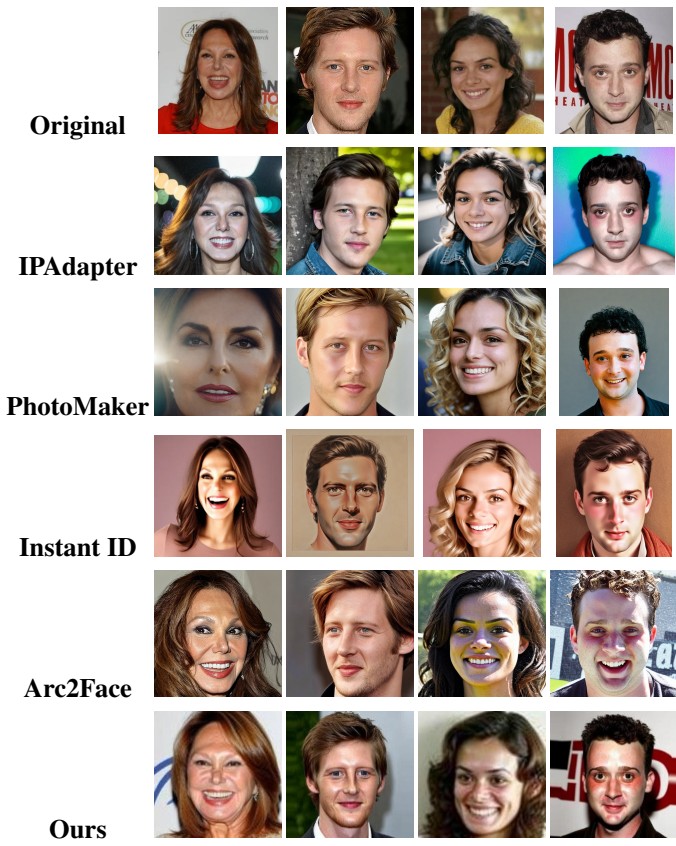

Figure 7: Identity preservation comparison across IPAdapter, PhotoMaker, Instant ID, Arc2Face, and Diff-ID.

## A.3   ABLATION STUDY FURTHER RESULTS

### A.3.1   MORPHING

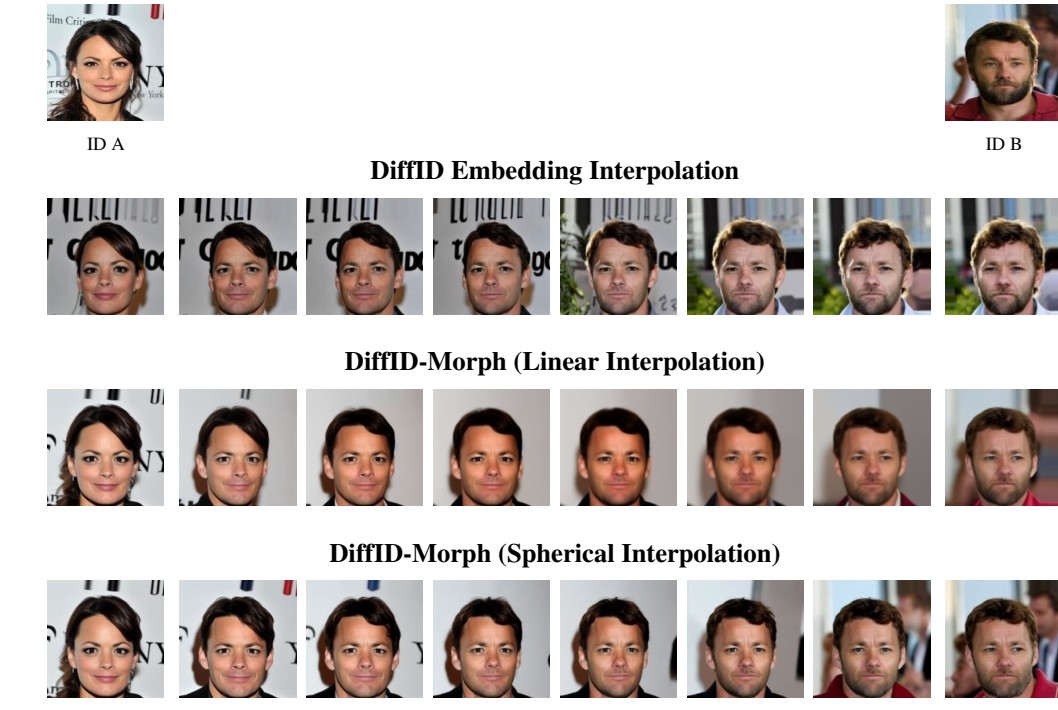

Figure 8: Comparison of morphing methods. Top-left (ID A) and top-right (ID B) are the original input identities. Middle shows interpolations via embedding, linear (Lerp), and spherical (Slerp) methods.

### A.3.2 VISUALIZATION OF DIFFERENT ADAPTER STRATEGIES

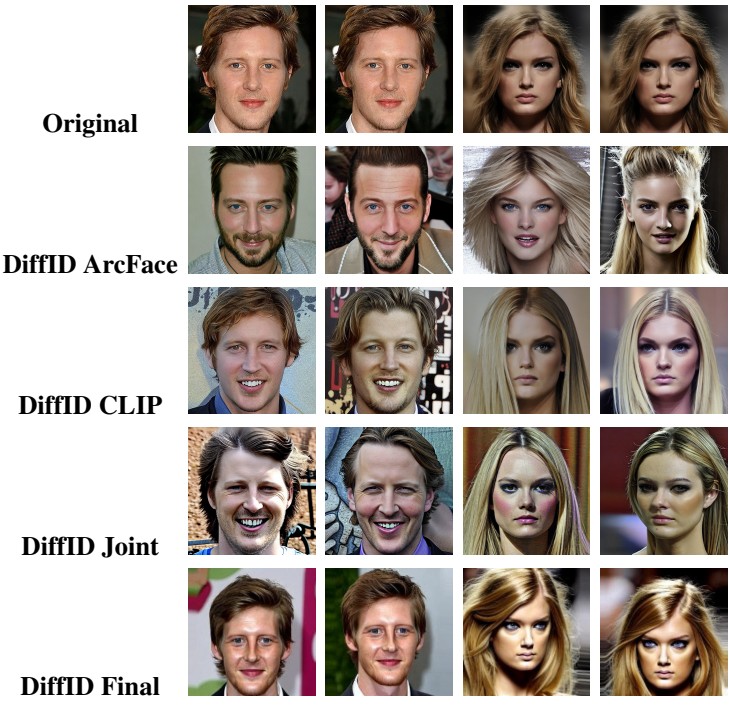

Figure 9: Adapter ablation study: identity preservation across variants.

### A.3.3 IMPACT OF IDENTITY LOSS GRAPH

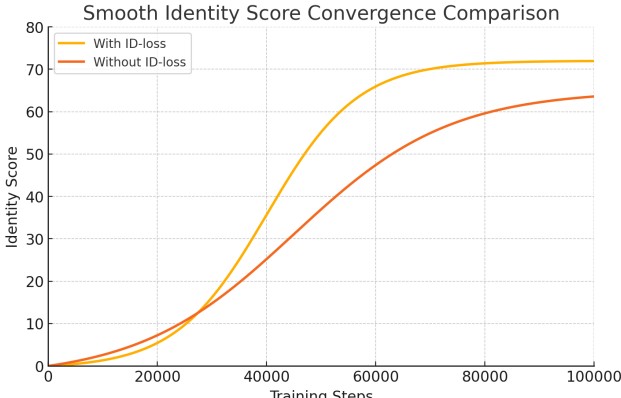

Figure 10: Identity score over timesteps, with and without the identity loss term.

