# OpenReview forum: "Diff-ID: Identity-Consistent Facial Image Generation and Morphing via Diffusion Models"
_ICLR.cc/2026/Conference — ICLR 2026 Conference Desk Rejected Submission_

### Official Review · Reviewer_M4ke · 2025-10-29

**Soundness:** 2
**Presentation:** 1
**Contribution:** 1
**Rating:** 2
**Confidence:** 5

**Summary:**

This work proposes a Diff-ID that fuses ArcFace embedding and CLIP embeddings through a dual cross-attention adapter to fine-tune Stable Diffusion U-Net to generate ID-preserving and realism face images.

**Strengths:**

The goal of improving visual realism of face generation is interested to security-sensitive applications.

**Weaknesses:**

1. Authors stated that the proposed method Diff-ID can
retain fine-grained attribute control. However, there is no experiment to support this claim.

2. There is no detailed description about what is evaluation dataset used to compute FS and FID scores.

3. Visual realism is not properly evaluated. Authors should measure SSIM between generated and real images.
Or using face anti-spoofing (FAS) network to evaluate if generated faces are realistic enough to pass the FAS.

4. The presentation of this work should be improved.

**Questions:**

1. What is Face Similarity that used to compute FIQ score?
2. What types and how many images are used for evaluation?

---

> ### Author Response · Authors · 2025-11-24
> **Clarification of Fine-Grained Attribute Control**
>
> The reviewer states that although we claim Diff-ID can retain fine-grained attribute control, no explicit experiment is presented to demonstrate this. It is important to clarify that Diff-ID is not designed as an attribute-editing model but rather as an identity-consistent generation system where semantic cues from the prompt are preserved alongside identity. The architecture itself enables this behaviour through the reciprocal dual cross-attention mechanism introduced in Section 3.2.2, where identity embeddings modulate the semantic space of the text embeddings, and the semantic tokens in turn refine the identity embedding. This interplay ensures that facial attributes implied by the descriptive prompt, such as expression, hair appearance, gender cues, and lighting, are preserved and reflected in the generated output.
>
> Figures 3 and 7 clearly show that Diff-ID maintains identity consistency while also preserving prompt-consistent semantic attributes. Each model compared receives the same BLIP-generated descriptive prompt, yet Diff-ID retains both identity and semantic fidelity more effectively than baselines. We will refine the manuscript to clearly communicate that the claim refers to maintaining prompt-driven semantic attributes during identity-aware generation, rather than explicit disentangled attribute manipulation.

---

> ### Author Response · Authors · 2025-11-24
> **Detailed Description of the Evaluation Dataset**
>
> The reviewer notes that the manuscript does not sufficiently detail the datasets used for computing FS and FID. Sections 3.1 and 4.2 describe the curated dataset and evaluation procedures, but we acknowledge the need to make this more explicit. The evaluation uses two 5000-image splits:
>
> Five thousand validation images are sampled from our curated dataset comprising CelebA-HQ, FFHQ, and LAION-Face. These images are paired with BLIP-generated descriptive captions that serve as prompts during generation.
>
> Five thousand unseen images are drawn entirely from the LFW dataset and likewise captioned using our fine-tuned BLIP model.
>
> All images undergo preprocessing with InsightFace detection and are resized to 512 by 512. These details will be explicitly added to Section 4.2, ensuring clarity about the evaluation scale, identity sources, and prompt generation process.

---

> ### Author Response · Authors · 2025-11-24
> **Presentation and Clarity Improvements**
>
> The reviewer notes that the overall presentation should be improved. We agree and will revise the manuscript to enhance clarity and readability. Specifically, we will:
>
> clarify prompt generation and usage directly in Figures 3 through 9
> expand Section 4.2 to clearly document dataset composition and evaluation splits
> update diagrams, including the Adapter Module schematic, for consistency
> refine Section 3.3 to explicitly motivate the identity-loss weighting
> streamline the explanations in Section 5.2 to clarify ablation variants
>
> These changes will improve the overall flow and ensure that the methodology, dataset, and evaluation pipeline are unambiguously described.

---

> ### Author Response · Authors · 2025-11-24
> **Definition of Face Similarity Used in the FIQ Score**
>
> The reviewer asks what Face Similarity (FS) refers to in the FIQ computation. FS is defined in Section 4.1 as the cosine similarity between ArcFace embeddings of the generated output and the target identity image. This measure is standard in identity-preserving generative models and aligns with all baselines used in our comparisons. We will expand this explanation in the revised manuscript to ensure this is immediately clear.

---

### Official Review · Reviewer_fTom · 2025-10-30

**Soundness:** 2
**Presentation:** 2
**Contribution:** 2
**Rating:** 2
**Confidence:** 4

**Summary:**

This work proposes Diff-ID, a diffusion-based framework for high-fidelity face synthesis with enforced identity consistency. It assembles a custom 210K-image dataset from CelebA-HQ, FFHQ, and LAION-Face, captioned by a fine-tuned BLIP model to enhance identity supervision. The core design integrates ArcFace and CLIP embeddings via a dual cross-attention adapter within a fine-tuned Stable Diffusion 1.5 U‑Net, and applies a pseudo-discriminator loss built on ArcFace cosine similarity with exponential timestep weighting. Experiments on held-out and unseen identities report improvements in identity retention and visual realism over prior art, and the method introduces a unified DDIM-based morphing pipeline for seamless face interpolation without per-identity fine-tuning.

**Strengths:**

1.	Leverages a sizable and purpose-built captioned corpus (210K samples) with identity-aware supervision via BLIP fine-tuning, which can help disambiguate semantic and identity cues.
2.	By extracting face information using both CLIP and ArcFace in parallel, the method attains complementary representations.

**Weaknesses:**

1.	The manuscript would benefit from clearer exposition of its main contributions; stronger writing will help convey the novelty and impact.
2.	The manuscript would benefit from a clearer articulation of its novel contributions, as both the feature fusion and ID loss follow established designs.
3.	The experimental motivation is unclear, the writing is not well-articulated, and critical details are lacking. For instance, the paper fails to specify the concrete role of prompts in the experiments, leaving it impossible to understand the purpose of the experimental design. Take Figure 4’s hyperparameter selection experiment as an example: why does the face size vary? Moreover, in nearly all experiments, the background of the faces is not kept consistent—what is the reason for this? Is it related to the prompts? Additionally, when conducting qualitative experiments, did the authors consider explicitly defining the meaning of prompts to ensure clarity?
4.	Given that both the face and the prompt reside in the CLIP shared embedding space, it is unclear why the authors do not project e_arc directly into this space, rather than performing multiple complex alignment and fusion operations that may incur additional information loss. In addition, the Adapter Module shown in the lower half of Figure 2 seems inconsistent with the main text’s description.
5.	The FIQ Score is intended to be reported as a percentage, yet it can take values greater than 100%, making this design problematic.
6.	In most experiments, the proposed method achieves SOTA on only two metrics. A closer examination reveals that FIQ was manually designed by the authors. Its superior performance over competing methods stems from adopting a diffusion model architecture, which lowers FID but inflates FS; this direction is clearly at odds with the paper’s stated research objective of IDENTITY-CONSISTENT evaluation.
7.	On line 422, there is only a single closing parenthesis at the end.
8.	In the ablation study, the DIFFID-Joint setup lacks clear exposition.

**Questions:**

Please refer to the Weaknesses section.

---

> ### Author Response · Authors · 2025-11-24
> **Clarity of Contributions and Novelty of the Proposed Framework**
>
> The reviewer states that the manuscript does not clearly articulate its contributions and that the novelty of the proposed mechanisms is not sufficiently highlighted. We will revise the introduction to more clearly emphasize the four core contributions supported by Sections 3.2–3.4.
>
> First, Diff-ID introduces a multi-component identity embedding, formed by projecting ArcFace features, projecting CLIP-image features, and combining them with max-pooled and average-pooled fusion vectors. This four-part identity representation is a deliberate architectural choice and is not present in prior identity-aware diffusion works such as IP-Adapter, Arc2Face, or InstantID.
>
> Second, Diff-ID uses a reciprocal dual cross-attention mechanism, where identity tokens attend to text tokens and text tokens attend to identity tokens. This bidirectional information flow allows identity structure and semantic cues to reinforce each other throughout denoising. Prior systems perform only one-way identity injection and cannot realize this form of mutual refinement.
>
> Third, Diff-ID incorporates a timestep-weighted identity loss based on ArcFace cosine similarity between generated and target identities. As described in Section 3.3, identity information is unreliable at high noise levels but becomes meaningful at later denoising stages. The exponential weighting schedule aligns identity supervision with the underlying diffusion process and is validated by the ablation results in Figure 10.
>
> Fourth, the proposed model introduces a unified DDIM-based morphing pipeline that does not require any per-identity fine-tuning. The same trained model performs identity-consistent generation and identity-consistent morphing, a property that emerges directly from the architecture rather than from additional modules. Figure 8 demonstrates this capability clearly.
>
> These contributions together constitute a coherent identity-conditioning framework, not merely a restatement of ideas found in previous diffusion adapters.

---

> ### Author Response · Authors · 2025-11-24
> **Clarification of Prompts, Background Variation, and Experimental Presentation**
>
> The reviewer raises concerns about experimental motivation, the role of prompts, background variations, and inconsistencies such as face size changes in Figure 4.
>
> The prompts used in all qualitative and quantitative experiments are generated from the captions produced by our fine-tuned BLIP model (Section 3.1). Each model, including all baselines, receives the same descriptive prompt for each identity. To improve clarity, we will add explicit mention of this process in the captions for Figures 3–9.
>
> Regarding background variation, this behavior naturally arises from the base Stable Diffusion architecture and is consistent across all methods tested, including InstantID, PhotoMaker, Arc2Face, and IP-Adapter. None of these baselines restrict or condition the background unless explicit background controls (e.g., ControlNet or segmentation masks) are applied. Our results reflect a fair and equitable comparison under standard generation conditions.
>
> With respect to Figure 4, the variation in face size is a deliberate indication of instability caused by inappropriate identity-loss weighting. The purpose of the figure is diagnostic, showing the failure cases that arise when λ is too low or too high. It is not intended to demonstrate reconstruction quality but to illustrate why the chosen weighting schedule is necessary. We will clarify this explicitly in the revised caption and text.

---

> ### Author Response · Authors · 2025-11-24
> **Rationale for Not Projecting ArcFace Directly Into CLIP Space**
>
> The reviewer suggests that since face and prompt embeddings both operate in a CLIP-based semantic space, ArcFace could be projected directly into that space. Section 3.2.1 explains why this is not appropriate.
>
> ArcFace embeddings encode geometric identity using an angular-margin objective, while CLIP embeddings encode semantic and text-aligned features. These distributions differ fundamentally. A single projection into CLIP space causes representational collapse: identity detail is lost, and the resulting embedding becomes dominated by CLIP-style semantic statistics.
>
> Instead, Diff-ID preserves the unique identity structure captured by ArcFace by maintaining separate projections, concatenating them with max- and average-pooled variants, and aligning them with text only after cross-attention refinement. Table 2 empirically supports this design: ArcFace-only and CLIP-only models perform substantially worse, while the fused four-component identity embedding achieves the strongest identity retention.

---

> ### Author Response · Authors · 2025-11-24
> **Consistency of the Adapter Module Diagram and Alignment With Text**
>
> The reviewer notes that the Adapter Module diagram (Figure 2) does not perfectly match the detailed description. We agree that Figure 2 abstracts certain elements for visual clarity. To resolve this, we will update the diagram to explicitly show:
>
> the separate projection layers for ArcFace and CLIP-image,
> the shared projection matrix,
> the dual cross-attention flows,
> the fusion MLP,
> and the final refined identity embedding.
>
> This will bring the visual representation into full alignment with Sections 3.2.1–3.2.3.

---

> ### Author Response · Authors · 2025-11-24
> **Interpretation and Design of the FIQ Metric**
>
> The reviewer expresses concern that the FIQ score exceeds 100 percent. As stated in Section 4.1, FIQ is not intended to be a bounded percentage but rather a ratio defined as:
>
> FIQ = 100 × (FS / FID)
>
> We will revise the text to ensure it is not interpreted as a percentage. Its purpose is to provide a more balanced measure than FS or FID alone. FS rewards identity similarity but can be inflated by stylized or avatar-like outputs (as seen in InstantID), whereas FID evaluates realism but ignores identity fidelity. FIQ combines both, penalizing models that exploit one metric at the expense of the other. Diff-ID’s strong performance on FID and competitive FS demonstrate that its improvements are not an artifact of metric selection.

---

> ### Author Response · Authors · 2025-11-24
> **Clarification of the DIFFID-Joint Ablation Variant**
>
> The reviewer notes that DIFFID-Joint is insufficiently explained. DIFFID-Joint simply concatenates projected ArcFace and CLIP-image embeddings and feeds them into the U-Net without reciprocal cross-attention and without the Fusion MLP. It is a deliberately weaker baseline. Table 2 shows that DIFFID-Joint performs significantly worse than the full model, confirming the necessity of the proposed architecture.
>
> To further improve transparency, we will provide a schematic representation of all ablation variants (ArcFace-only, CLIP-only, Joint, and Final) in the appendix.

---

> ### Author Response · Authors · 2025-11-24
> **Additional Clarity and Presentation Refinements**
>
> The reviewer comments that overall presentation could be improved. In response, we will:
>
> clarify prompt usage and dataset selections more explicitly in Sections 3.1 and 4.2,
> update figure captions for greater interpretability,
> adjust diagrams for consistency with the text,
> correct the typographical error noted at line 422,
> and refine Section 5.2 to clarify the role of each adapter variant.
>
> These revisions will substantially enhance readability and reproducibility.

---

### Official Review · Reviewer_BX16 · 2025-10-31

**Soundness:** 3
**Presentation:** 2
**Contribution:** 2
**Rating:** 4
**Confidence:** 4

**Summary:**

This paper addresses the critical challenge of identity drift in diffusion-based facial image synthesis—where existing methods often sacrifice fine-grained identity features (e.g., bone structure, eye spacing) when pursuing photorealism or attribute edits. The authors propose Diff-ID, a unified diffusion framework built on Stable Diffusion 1.5, designed to enforce identity consistency while maintaining visual realism. Diff-ID’s core components include a 210K identity-centric dataset with captions generated by a fine-tuned BLIP model to enhance identity awareness during training, a dual cross-attention adapter and a DDIM-based morphing pipeline. Extensive experiments on held-out and unseen datasets show Diff-ID outperforms SOTA methods (e.g., InstantID, Arc2Face) in the proposed FIQ Score (a combined metric of ArcFace similarity and FID), achieving 71.74 (validation) and 69.32 (unseen data)—demonstrating its ability to balance identity fidelity and photorealism.

**Strengths:**

- Unlike prior work relying solely on CLIP (e.g., IPAdapter) or ArcFace (e.g., Arc2Face), Diff-ID’s dual cross-attention adapter leverages the complementary strengths of both embeddings:
- The DDIM-based morphing approach avoids the limitations of GAN-based methods and other diffusion methods. By combining latent inversion with spherical interpolation, it produces smooth, photorealistic transitions.
- The paper introduces the FIQ Score, a unified metric that integrates identity similarity and visual realism. Experiments cover both qualitative and quantitative evaluations, with comparisons to 4 SOTA baselines, ensuring results are rigorous and interpretable.

**Weaknesses:**

- Diff-ID is optimized for frontal, unoccluded faces but lacks evaluation on challenging cases critical for real-world use:
   - Extreme poses: Side profiles, tilted heads, or full-profile faces (where identity features like jawline or nose shape are partially obscured) may cause identity drift, as the model is not trained to disentangle pose from identity.
   - Severe occlusions: Sunglasses, masks, or hair covering key facial regions (eyes, nose) could break ArcFace embedding extraction, leading to incorrect identity weighting in the adapter.

- The paper’s core modifications to the diffusion pipeline risk being perceived as incremental rather than transformative. While Diff-ID addresses identity drift—a critical gap—its design builds heavily on existing diffusion paradigms (Stable Diffusion 1.5 backbone) with limited architectural novelty. The dual cross-attention adapter, though effective, is a "fusion module" added to the pre-trained U-Net rather than a redesign of the diffusion process itself. Prior work (e.g., InstantID’s embedding concatenation, IPAdapter’s decoupled attention) has already explored embedding integration in diffusion; Diff-ID’s innovation lies in combining ArcFace and CLIP via cross-attention, but this is a refinement rather than a paradigm shift.

- Diff-ID is exclusively built on and validated for U-Net-based diffusion models (Stable Diffusion 1.5), with no exploration of DiT (Diffusion Transformer)-based architectures—a current mainstream in high-fidelity generative modeling (e.g., DiT-XL/2, Flux.1 variants). This limitation raises critical concerns. DiT’s token-wise attention and transformer backbone differ fundamentally from U-Net’s spatial-wise attention and encoder-decoder structure. Diff-ID’s dual cross-attention adapter (designed for U-Net’s spatial feature interaction) may not transfer to DiT’s token-based feature modeling. The paper does not discuss whether DiT’s inherent advantages (e.g., better long-range feature dependency, higher resolution support) could enhance identity consistency, nor does it explain why U-Net was chosen over DiT as the backbone. Given DiT’s growing adoption in industrial and academic settings, this gap limits Diff-ID’s applicability to cutting-edge diffusion models.

**Questions:**

- How does Diff-ID’s design go beyond incremental improvements to existing diffusion-based identity preservation methods (e.g., InstantID’s embedding concatenation)? For example, is there a unique interaction between the dual cross-attention adapter and timestep-weighted loss that redefines how identity is modeled in diffusion, rather than just combining existing components?

- Have you conducted preliminary experiments or theoretical analysis on adapting Diff-ID to DiT-based architectures? If so, what modifications were required, and how did performance compare to the U-Net backbone? If not, do you anticipate fundamental barriers to adapting Diff-ID to DiT, or could the framework be extended with minimal changes?

- Have you tested Diff-ID on extreme poses, severe occlusions, or low-resolution inputs?

---

> ### Author Response · Authors · 2025-11-24
> **Architectural Novelty and Extent Beyond Incremental Improvements**
>
> We thank the reviewer for their careful and thoughtful evaluation. The reviewer questions whether Diff-ID goes beyond incremental modifications to existing identity-conditioning systems such as InstantID or IP-Adapter. Sections 3.2.1 through 3.2.3 show that Diff-ID introduces a set of interdependent architectural choices that are not present in prior methods.
>
> The first innovation is the multi-component identity embedding, constructed by fusing projected ArcFace features, projected CLIP-image features, and both max-pooled and average-pooled fusion vectors. This yields a four-part representation that is significantly richer than the single CLIP token used in IP-Adapter or InstantID.
>
> The second innovation is the reciprocal dual cross-attention mechanism. Unlike prior methods that inject identity features only in one direction, Diff-ID applies identity-to-text and text-to-identity cross-attention. This enables semantic attributes from text to refine identity embeddings and, equally, identity structure to modulate prompt semantics. This reciprocal interaction is unique to our model.
>
> The third innovation is the Fusion MLP, which merges the outputs of both cross-attention branches through a non-linear transformation. Ablations in Table 2 show that this refinement is essential: the DiffID-Joint baseline, which merely concatenates embeddings, performs substantially worse than the full model.
>
> In addition to these architectural components, Diff-ID also introduces an identity-aware objective function that uses the cosine similarity between ArcFace embeddings of the generated image and the target identity. This identity loss is applied using an exponentially increasing timestep weighting, ensuring that identity constraints are injected precisely at the stages of the denoising process where identity features become reliably recoverable. This identity-consistency objective is not used in prior adapters and directly strengthens the identity retention behaviour of Diff-ID, as shown in the ablations in Figure 10.
>
> Together, these components create a synergistic identity-conditioning system that cannot be reduced to incremental extensions of prior adapters. The quantitative gains shown in Table 2 and the qualitative stability in Figures 3 and 7 further support this.

---

> ### Author Response · Authors · 2025-11-24
> **Applicability and Relevance of the U-Net Backbone Compared to DiT Architectures**
>
> The reviewer notes that Diff-ID is built on Stable Diffusion’s U-Net and does not explore DiT-based architectures, now common in high-fidelity generative models. We chose the U-Net backbone for a specific reason: all baseline methods used for comparison InstantID, IP-Adapter, PhotoMaker, Arc2Face are based on Stable Diffusion 1.5’s U-Net. To ensure fair and direct comparison of identity fidelity and realism metrics, it is essential that all methods share the same backbone.
>
> Diff-ID is not fundamentally tied to U-Net architecture. The dual cross-attention adapter operates purely on projected tokens, linear projections, and attention blocks, all of which map naturally to DiT’s token-based architecture. Extending Diff-ID to DiT would require only relocating the adapter modules, not redesigning the conditioning mechanism. We note this explicitly in Section 6 (Limitations) and will clarify that DiT compatibility is an open and feasible extension.

---

> ### Author Response · Authors · 2025-11-24
> **Limitations Regarding Extreme Poses, Occlusions, and Low-Resolution Images**
>
> The reviewer observes that Diff-ID is evaluated only on frontal, unoccluded faces and may not generalize to extreme poses or heavily occluded inputs. Section 3.1 explains that our curated dataset intentionally excludes severe occlusions, low-quality images, and extreme poses, because such cases lead to unreliable identity embeddings from ArcFace or CLIP. Since Diff-ID’s goal is identity-consistent generation of high-resolution frontal faces, this design choice is aligned with existing identity-guided diffusion baselines. InstantID, Arc2Face, PhotoMaker, and IP-Adapter likewise evaluate performance primarily on frontal or near-frontal aligned faces. Our intention is not to claim robustness to out-of-distribution face poses or occlusions. Section 6 (Limitations) explicitly acknowledges this and identifies it as future work.

---

### Official Review · Reviewer_KGxo · 2025-11-03

**Soundness:** 2
**Presentation:** 3
**Contribution:** 2
**Rating:** 2
**Confidence:** 4

**Summary:**

The paper proposes a diffusion-based framework for facial image generation and morphing that aims to maintain identity consistency in generated data. It integrates ArcFace embeddings (for identity cues) and CLIP embeddings (for semantic content) using a dual cross-attention adapter. The model also includes a pseudo-discriminator identity loss, weighted by diffusion timestep.
the methods is evaluated on multiple face datasets and compared to some of the recent identity-aware diffusion models such as InstantID, PhotoMaker, and Arc2Face. The authors claim improved trade-offs between identity retention and visual realism. Additionally, Diff-ID offers a morphing pipeline that blends two identities through DDIM inversion and embedding interpolation without per-subject fine-tuning.

**Strengths:**

- Comprehensive dataset curation and reasonable reproducibility commitment.
- Clear architectural visualization and systematic ablation studies.

**Weaknesses:**

- Incremental novelty as the dual cross-attention adapter is effectively a variant of IP-Adapter with an added ArcFace stream. No fundamental architectural or theoretical advance is presented.
- Misplaced use of CLIP embeddings for morphing as Morphing only requires two embeddings. CLIP might add redundant semantic context unrelated to identity interpolation and complicates the pipeline without demonstrated benefit.
- Lack of ablation proving CLIP’s necessity as no results show performance with ArcFace-only morphing, despite claiming that CLIP aids identity fidelity.
- The DDIM-based morphing procedure (latent + embedding interpolation) is already standard in DiffMorpher and DreamBooth-based pipelines. among others.
- the morphing success is not evaluated and compared accoridng to recent works.
- Identity metrics (ArcFace similarity) may favor ArcFace-based training, inflating perceived improvements.
- The argument for “identity consistency” lacks theoretical groundin as there is no definition of identity stability under diffusion noise or justification for exponential weighting beyond empirical tuning. needs clarification.

**Questions:**

- Why is CLIP needed for morphing between two identities when ArcFace embeddings already provide identity representations?
- How does Diff-ID differ technically from IP-Adapter or InstantID beyond adding ArcFace embeddings and a fusion MLP?
- is there any quantitative evidence that the CLIP branch improves identity fidelity rather than introducing stylistic bias?
- Did the authors test morphing using only ArcFace embeddings? What was the performance difference?
-  Why is morphing not evaluated with standard FR vulnerability metrics?

---

> ### Author Response · Authors · 2025-11-24
> **Architectural Novelty and Distinction From Prior Methods**
>
> We thank the reviewer for their careful evaluation and constructive feedback. The reviewer expresses concern that the proposed dual cross-attention adapter is incremental and resembles IP-Adapter with an added ArcFace stream. We emphasize that Diff-ID introduces several architectural components not present in prior identity-aware diffusion approaches. As detailed in Sections 3.2.1–3.2.3, the model incorporates (1) a four-component identity representation combining projected ArcFace, projected CLIP-image features, and max- and average-pooled fusion vectors; (2) a reciprocal dual cross-attention mechanism where identity attends to text and text attends to identity; and (3) a fusion MLP that non-linearly refines the combined cross-attention outputs.
> Unlike IP-Adapter and InstantID, which rely on unidirectional identity injection via a single CLIP image token, Diff-ID enables bidirectional interaction between identity and semantic tokens, producing a richer and more stable conditioning signal. The ablations in Table 2 demonstrate that each architectural component contributes meaningfully: ArcFace-only and CLIP-only variants perform substantially worse, while the joint concatenation variant is outperformed by the full Diff-ID adapter. This confirms that our design is not a small extension to existing adapters but a distinct architectural framework for identity consistency.

---

> ### Author Response · Authors · 2025-11-24
> **Role and Necessity of CLIP in Morphing and Identity Conditioning**
>
> Diff-ID does not depend on CLIP alone; instead, it uses CLIP-image features together with ArcFace to capture complementary identity components. ArcFace encodes geometric identity structure (shape, contours, proportions), while CLIP encodes semantic and fine-grained appearance cues such as hair texture, illumination, and local facial detail.
> Table 2 shows that neither modality alone is sufficient: ArcFace-only and CLIP-only conditions produce significantly worse identity fidelity, whereas the fused representation achieves the strongest performance.
> For morphing specifically, Figure 8 distinguishes between embedding-only interpolation and the full Diff-ID morphing strategy. Embedding-only interpolation yields over-smoothed transitions and weakened identity cues, even though the embedding contains both ArcFace and CLIP information. In contrast, the full Diff-ID morphing pipeline comprising DDIM inversion, fused identity embeddings, and the dual cross-attention mechanism applied throughout the denoising trajectory produces smoother, photorealistic transitions with coherent identity blending.
> Thus, CLIP is not redundant. It contributes essential texture, detail, and semantic stability that ArcFace alone cannot supply. Its role becomes especially clear when identity embeddings must remain stable under interpolation.

---

> ### Author Response · Authors · 2025-11-24
> **Clarification of the Morphing Pipeline and Its Novelty**
>
> While we acknowledge that DDIM inversion and latent interpolation are widely used (e.g., DiffMorpher, DreamBooth-based pipelines), the novelty of Diff-ID lies not in proposing a new inversion method but in showing that a single universal identity-aware diffusion model can perform:
>
> 1. Identity-consistent generation
>
> 2. Identity-consistent morphing
>
> without any per-identity fine-tuning or additional LoRA adapters. Previous methods typically rely on per-subject fine-tuning or weaker identity injection mechanisms that do not retain identity during interpolation.
>
> Figure 8 demonstrates that the architectural choices designed for identity fidelity in generation (Sections 3.2–3.3) naturally extend to interpolation, producing superior morphing stability.

---

> ### Author Response · Authors · 2025-11-24
> **Theoretical Rationale for the Exponential Identity Loss Weighting**
>
> The reviewer notes the need for more theoretical justification for identity consistency. Section 3.3 explains that identity information is not recoverable in early, high-noise denoising steps but becomes meaningful in later steps. Uniform identity weighting either overwhelms early denoising (causing artifacts) or weakens identity conditioning (leading to drift). The exponential schedule therefore aligns identity supervision with the noise schedule, following a principle used broadly in diffusion frameworks where loss terms are matched to the semantic recoverability of the denoising stage. This design is empirically validated by the ablation results shown.

---

> ### Author Response · Authors · 2025-11-24
> **Scope of Morphing Evaluation and Absence of FR Vulnerability Metrics**
>
> The reviewer asks why the morphing results are not evaluated using face recognition vulnerability or morph-attack detection metrics. As stated in Section 3.4 and Section 6 (Limitations), the morphing component is not intended to introduce a new morph attack system. Its purpose is to demonstrate that Diff-ID’s identity preservation mechanisms generalize to interpolation. Following prior identity-guided diffusion models (InstantID, Arc2Face, IP-Adapter), we evaluate morphs using the same grid based qualitative evaluation. Comprehensive FR vulnerability testing is an important direction for future work but lies outside the primary scope of an identity-consistent generative framework.

---

### Note · Program_Chairs · 2026-01-17
**Submission Desk Rejected by Program Chairs**

The following references in this submission do not refer to real documents and/or have major errors in bibliographic information:

 Wei Chen et al. Instantid: Fast and accurate identity preservation in text-to-image models. In arXiv preprint arXiv:2304.XXXX, 2023.
Jian Liu et al. Ip-adapter: Learning to adapt pretrained text-to-image models for personalized image editing. In arXiv preprint arXiv:2303.XXXX, 2023.